# Polyphonic Music Composition: An Adversarial Inverse Reinforcement Learning Approach

## Abstract

Most recent approaches to automatic music harmony composition adopt deep supervised learning to train a model using a set of human composed songs as training data. However, these approaches suffer from inherent limitations from the chosen deep learning models which may lead to unpleasing harmonies. This paper explores an alternative approach to harmony composition using a combination of novel Deep Supervised Learning, Deep Reinforcement Learning and Inverse Reinforcement Learning techniques. In this novel approach, our model selects the next chord in the composition(action) based on the previous notes(states), therefore allowing us to model harmony composition as a reinforcement learning problem in which we look to maximize an overall accumulated reward. However, designing an appropriate reward function is known to be a very tricky and difficult process. To overcome this problem we propose learning a reward function from a set of human-composed tracks using Adversarial Inverse Reinforcement Learning. We start by training a Bi-axial LSTM model using supervised learning and improve upon it by tuning it using Deep Q-learning. Instead of using GANs to generate a similar music composition to human compositions directly, we adopt GANs to learn the reward function of the music trajectories from human compositions. We then combine the learned reward function with a reward based on music theory rules to improve the generation of the model trained by supervised learning. The results show improvement over a pre-trained model without training with reinforcement learning with respect to a set of objective metrics and preference from subjective user evaluation.

## 1 Introduction

Automatic music composition usages can vary from continuous generation of copyright free music for use in media to inspiration tools for musicians. With recent advances of neural networks and with the availability of large set of big data on digitized music scores composed by humans, the trend of automatic music composition has shifted to learning an embedded model from big data and then generating new music based on the learned model. Computation models used to generate music need to be able to find a balance between adhering to music theory rules and exploring new chord progressions in order to generate interesting and pleasing music harmonies. If the model sticks too much to music theory rules, the composed music can be harmonious but may also turn out to be dull and uninteresting. On the other hand, if the model only explores new chord progressions while ignoring music theory rules, the composed music may include inharmonious chord progressions.

If we look at music composition problem from the symbolic level, it can be viewed as generating a sequence of music chords, in which selecting each chord's vector representation is equivalent to an action. A chord in the composition is a set of notes played at the same time, with each note being represented by a pitch value. Reinforcement Learning (RL) can be used to formulate the creative problem of music composition by viewing it as a sequence of chord-selecting actions. The idea for music generation based on RL can be viewed as assigning a reward on the selected notes/chords while the music composition agent is deemed to maximize the accumulated reward by selecting a sequence of most appropriate chords to generate the piece of music. However, in this formulation designing a good reward signal can be a critical and challenging task. In order to overcome this limitation, instead of only manually and subjectively assigning the reward for each chord selection

action, we propose using Inverse Reinforcement Learning (IRL) to learn a reward function based on the optimal actions taken from the composed master pieces of human expert musicians.

Therefore, in this paper, we design a novel model that is able to learn a reward function from human composed songs and then use that reward function in a RL model that tunes a pre-trained model to improve the generation of piano harmonies. The model has three main training phases and uses the human composed songs as training data. First, we use deep supervised learning to pre-train a model to learn from the training human composed music data, then we learn a reward function from the same training data using IRL and finally we tune the pre-trained model with RL using the combination of learned reward function from IRL and a supporting music theory reward.

## 2 RELATED WORKS

Music Generation with deep learning models has been explored extensively from various perspectives. Music generation can be tackled from the audio signal waveform approach (van den Oord et al., 2016) or from the symbolic representation of music score that can be later transformed to the audio file. The latter approach has proven to be faster for generation purposes compared to generating the audio waveform directly, albeit with a trade-off in the range of sounds that are possible to generate. Recurrent Neural Networks (RNN) (Sturm et al., 2016), Convolutional Neural Networks (CNN) (Yang et al., 2017) and Transformers (Huang et al., 2018) are the most common architectures used for music generation. Generative Adversarial Networks (GAN) (Goodfellow et al., 2014) based approaches have gained popularity recently due to their success in the image generation and natural language Q/A fields, with some authors also finding success in modeling sequence generation of music (Guimaraes et al., 2017). GANs are famous for being able to generate pieces of music similar to given illustrative examples using a generator to compose music tracks and a discriminator to distinguish between the real compositions (the given illustrative examples) and the fake ones (the tracks composed by the generator). However, when using GANs to generate music compositions directly, it tends to generate pieces of music too similar to the given illustrated examples that lack novelty and creativity.

As RL is commonly used to learn a policy function that can maximize the accumulated reward based on actions carried out under a policy, it can be used in many different applications such as robot action sequences learning along with many others. Applying RL for music generation is a relatively new idea. The RL Tuner architecture proposed by the Google Magenta Team (Jaques et al., 2017) is one of the earliest works that combines a deep supervised learning model with deep RL for additional tuning. This model could only generate monophonic melody, but it was later expanded upon by Kotecha (Kotecha, 2018a) on his Bach2Bach model that could generate polyphonic harmonies using the same ideas from the RL Tuner architecture. Bach2Bach's tuning is only done with monophonic notes though, so even if the original model is capable of modeling polyphonic chords, the action space used for RL tuning is limited to only monophonic notes. Our model goes a step further by using polyphonic chords as the actions when tuning the reward. Another of Bach2Bach model's limitations is the difficulty of manually designing a reward signal. This is an important limitation because the tuning is highly dependent on the reward signal. IRL can be used to solve this, as it can help to learn a reward function from a set of expert demonstrations. One previous paper explored the use of IRL in the music domain (Messer & Ranchod, 2014). This paper used a small dataset with non-scalable RL and IRL algorithms. Furthermore, their approach only generated monophonic melodies. To our knowledge our approach is the first deep learning model based on IRL and adversarial learning that is able to generate polyphonic harmonies.

## 3 BACKGROUND

### 3.1 BI-AXIAL LSTM

Daniel Johnson (Johnson, 2017) introduced a new architecture called Bi-axial LSTM based on LSTM cells purposely built for music composition. This architecture takes inspiration from CNNs and is composed of 4 layers of LSTM cells with 2 layers having recurrent connections on a time axis and 2 layers having recurrent connections on a note axis. The input of the Bi-axial model represents both the polyphonic music nature as well as the difference in the articulation of a note.

## 3.2 Deep Q-Learning

One of the most recent breakthroughs in RL came with the successful implementation of a deep learning model to learn a policy from a high-dimensional input by the Deep Mind team (Mnih et al., 2013). Their approach, known as Deep Q-learning, uses a CNN trained with a variant of Q-learning that outputs the state-action value function. The main idea of this approach is to use a neural network to approximate the state-action value function normally represented in a tabular way in normal Q-learning. The state-action value function is now represented as a function of the state $s$, action $a$, and weights $\theta$: $Q(s, a; \theta)$. In order to stabilize and improve learning some additional techniques like experience replay (Lin, 1993) and using an additional Target Q-Network for estimating the target Q-value are used.

## 3.3 Adversarial Inverse Reinforcement Learning

IRL's goal is to learn a reward function that can best explain observed expert behaviors. IRL (followed by an RL algorithm to learn a policy) can sometimes be more effective at learning from expert behaviors than supervised learning, as argued by Abbeel and Ng (Abbeel & Ng, 2004). The reward function learned by IRL algorithms is commonly represented as $r_\theta(s) = \theta^T \cdot \phi(s)$, where $\theta$ is the learned weights and $\phi$ is a vector of features used to represent the state $s$.

Fu proposed a new model-free scalable IRL algorithm based on an adversarial reward learning formulation called Adversarial Inverse Reinforcement Learning (AIRL) (Fu et al., 2017). Fu uses concepts from Maximum Entropy IRL (Ziebart et al., 2008) and Generative Adversarial Network Guided Cost Learning (GAN-GCL) (Finn et al., 2016) to solve the IRl problem. In the AIRL algorithm the policy $\pi$ works as the Generator and is trained to fool a discriminator by generating trajectories that eventually become similar to the expert trajectories. A special structure enforced in the discriminator allows the reward function $r_\theta$ to be recovered from the discriminator. The AIRL algorithm alternates between training a discriminator to classify between expert trajectories and learner policy samples, and updating the policy to try to confuse the discriminator.

## 4 Approach

Our approach is to tune a pre-trained polyphonic generation model, in this case a Bi-axial LSTM, using Deep Q-Learning with the reward function obtained from the AIRL algorithm combined with a supporting music theory reward to generate harmonies that maximize the total reward. To accomplish this task, we proposed a design based on the RL Tuner (Jaques et al., 2017) introduced by the Google Magenta Team and later expanded upon by Kotecha's Bach2Bach approach (Kotecha, 2018a). Just like Bach2Bach, our approach trains a polyphonic harmony model and later tunes it using RL. The novelty of our approach is the use of an AIRL learned reward during the tuning process and the use of polyphonic chords as actions when training with RL.

Our design is based on 3 main training phases:

1. A Bi-axial LSTM model is trained on large data samples. This model will be referred as the pre-trained model.

2. A state-action reward function is learned using AIRL from the same data set used to train the Bi-axial model.

3. A Deep-Q Network is used for tuning the pre-trained Bi-axial model using a combination of the learned reward function from training phase 2 and a supporting music theory reward.

In order to incorporate some music theory rules and penalize unwanted behavior we also incorporated some music theory rewards into the final reward function. Figure 1 shows all of the three training phases and the relations between them:

### 4.1 Reward function extraction using AIRL

We use the AIRL algorithm from Fu (Fu et al., 2017) to learn a reward function from our training data. In order to apply the AIRL algorithm we need to model music generation as a RL problem. We

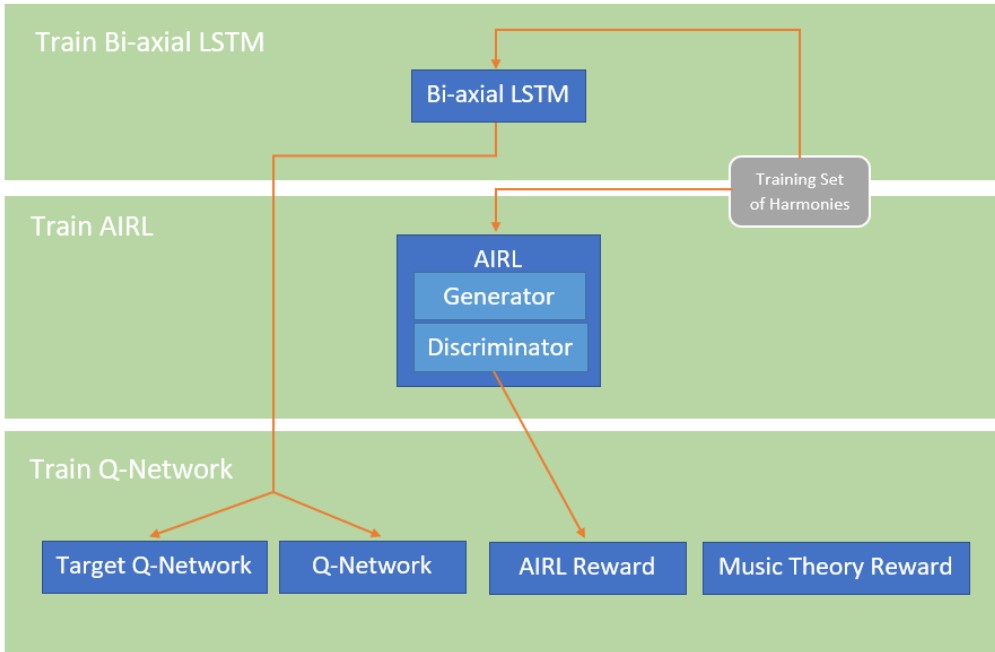

Figure 1: System overview

define an action as a new chord being played, and the state is represented by the previous chord. The midi fragments can be transformed into pianorolls, with each pianoroll representing one trajectory.

Our implementation uses the state-action variation of the discriminator proposed by Fu (Fu et al., 2017). Having a total of 44 possible notes that can be played at the same time and with our model being polyphonic it is clear that both the state space and action space are too big to model with discrete values (the total number of possible actions and states is $2^{48}$ each). Policy gradient methods offer a way to deal with both continuous spaces and continuous actions and are needed to model polyphonic harmony generation as a RL problem. These methods learn the statistics of a probability distribution from which the action is then sampled instead of computing the probabilities for each of the possible actions.

Following the AIRL algorithm, we need a stochastic parameterized policy that acts as a generator and a parameterized reward function that is part of the discriminator. The policy can be implemented using a Gaussian Policy, where the mean and standard deviation are the learnable parameters and are approximated with a basic Multilayer Perceptron (MLP) network. The reward function inside the discriminator is also approximated with a MLP network.

During each training iteration the policy generates multiple trajectories of chords. A set of state-action pairs is then sampled from the generated trajectories and combined with a set of state-action pairs sampled from the actual song fragments, which are considered the expert trajectories. This set containing half state-action pairs from the generated trajectories and half state-action pairs from the expert trajectories is used to train the discriminator to classify between the expert and generated trajectories. The reward function can be obtained thanks to the special structure imposed on the discriminator and updating the discriminator can be seen as updating the reward function. With this updated reward function, we can proceed to train the policy $\pi$ using a policy gradient method.

### 4.2 Tuning using a Deep Q-Network

A Deep Q-Network and a Target Q-Network are initialized with the weights of the pre-trained Bi-axial LSTM. The Q-Network is used to obtain the next action based on the previous state. The Target Q-Network is used to estimate the target, which in Q-learning is the reward obtained for the current action plus the maximum possible reward that can be obtained during the next state if

that action is selected. This Target Q-Network is slowly updated to be similar to the Q-Network. A reward calculated from a combination of the reward function learned from AIRL and some manually defined music theory rewards from the music theory reward module is obtained for every state-action pair.

The music theory reward module gives positive rewards based on if the action selected fulfills the conditions that we consider as following music theory rules or gives negative rewards if the selected action violates what we consider as a good harmonic chord progression in terms of music theory.

The music theory reward is composed of 4 different components based on the music theory rules implemented by the Google Magenta team in their RL Tuner (Jaques et al., 2017), the 4 of them are combined in the final music theory reward with each component having an equal weight for simplicity.

1. Harmony Reward: reward for continuous notes part of a scale or chords part of common chord progressions

2. Repeated Motif Reward: reward for repeating motifs in the composition

3. Chords in Composition Reward: reward for using chords that have appeared in the composition before

4. Chord Repetition Reward: reward that penalizes continuous repetition of chords or silence

Both the AIRL reward and music theory reward are normalized so their values are always between 0 and 1, this is to make it easier to assign more weight to one or the other. Our approach controls the weights using a constant $c$ to determine the weight of each reward. The combined total reward function for state $s$ and action $a$ is Eq. 1.

$$r_{total}(s,a) = c * r_{mt}(s,a) + (1-c) * r_{airl}(s,a) \tag{1}$$

Based on the Deep Q-Network loss function the final loss function becomes:

$$L(\theta) = \mathbb{E}_{\beta}\left[ (r_{total}(s,a) + \gamma max_{a'}Q(s',a';\theta^{-}) - Q(s,a;\theta))^2 \right] \tag{2}$$

The next action is obtained using an epsilon-greedy policy, allowing for exploration of the action space. If exploration is not performed the next chord is obtained by feeding the previous chord to the model and using the output as the next chord. So, for every chord in the composition (or state) a corresponding action is selected either by random exploration or by getting the next chord according to the model. Each state-action pair is saved in an experience replay buffer used for training. A random batch of samples is obtained from this experience replay buffer during training to be used as the states and actions in Eq. 2.

The Bi-axial LSTM architecture uses a final layer with its length being equal to the number of possible notes in the composition. This final layer uses a sigmoid activation function for each of the possible notes, learning the individual probability between 0 and 1 of each note being played while not being mutually exclusive. When selecting a new note to be played, a Bernoulli distribution is used to sample from each individual probability. This presents a problem for the loss function in Deep Q-learning, because when a state-action value is increased (when that specific state-action pairs receives a good reward) the other values that were not selected do not change. This leads to all values eventually having a high probability and the next chord being selected ends up as a combination of almost all possible notes. In order to overcome this problem, we add an extra softmax activation after the output layer of the Bi-axial LSTM used as a Q-Network before performing the back-propagation on the Deep Q-Network loss function. The $Q(s,a,\theta)$ in the Deep Q-Network loss function (Eq. 2) is calculated by multiplying the values from the added softmax activation in the Q-Network by the action selected. This multiplication works like a filter and is done to select the probabilities of the notes that were played in the action selected, because the action selected will contain zeros in the notes not selected and ones in the notes selected. $Q(s,a,\theta)$ is the mean of these selected probabilities. $max_{a'}Q(s',a',\theta^{-})$ is calculated by obtaining the highest probability from the Target Q-Network probabilities after feeding the next state to the Target Q-Network. The whole architecture of the RL tuning can be observed in Figure 2.

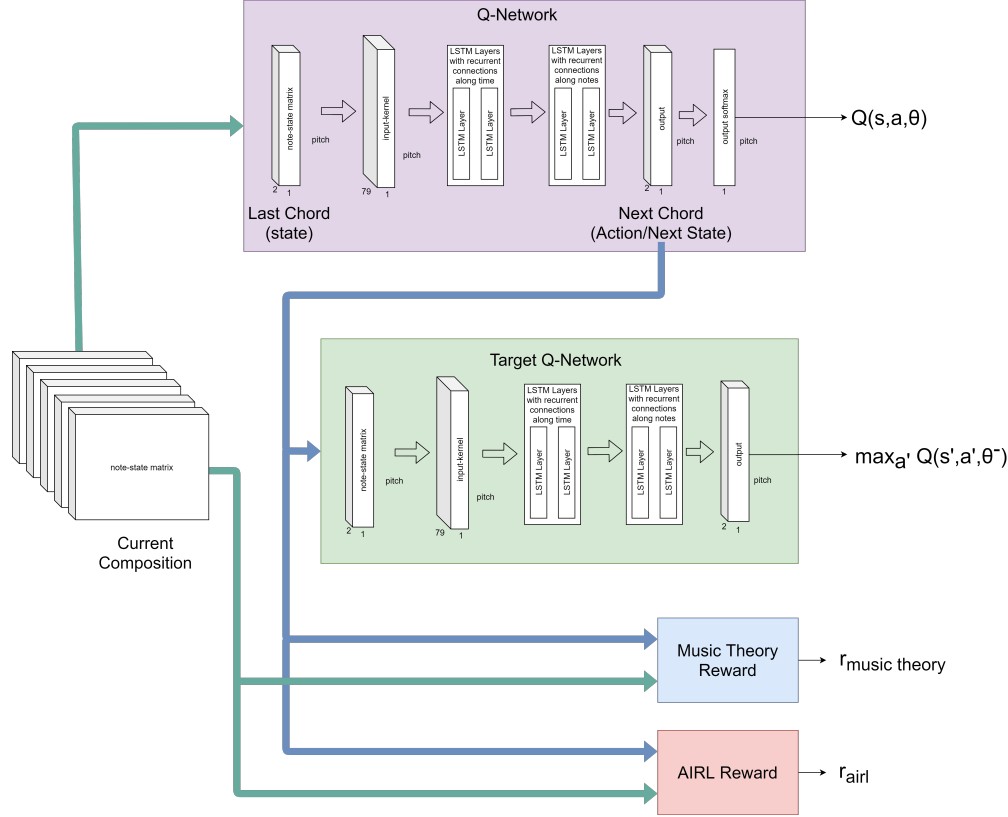

Figure 2: Reinforcement learning tuning architecture

## 5 EXPERIMENTS

The code of the deep learning models for experiments was implemented in python 3.6 under the tensorflow framework version 1.15. The code is extended from three major open sources. The code for Daniel Johnson's Bi-axial LSTM is based on the implementation by Nikhil Kotecha (Kotecha, 2018b). The code for the AIRL is based on the implementation by Justin Fu (Fu, 2018). The code for the deep RL training is based on the implementation done by Google Magenta Team (Team, 2020).

We chose the Lahk Pianoroll Dataset (Dong et al., 2018) (Raffel, 2016) for training. This dataset contains 174,154 multitrack pianorolls derived from the Lakh MIDI Dataset. The authors also provide some subsets of the dataset, the one we used is called the LPD-5 cleansed dataset, in which all the tracks are merged into five common categories: Drums, Piano, Guitar, Bass and Strings. This dataset was chosen because it provides a good amount of songs with diversity, having songs from all genres with a 4/4 time signature. Since our objective is not to generate a multi-track piece we only considered the songs with piano tracks. After filtering out the songs with no piano tracks we applied some extra pre-processing to the tracks. The final dataset used for training consists of 30 second fragments of the pre-processed tracks, with a total of 13,060 fragments. Using the whole range of midi pitch results in very sparse matrices, so we limited the range to 44 notes (pitch values from 36 to 80 in the MIDI file).

The Bi-axial LSTM consists of 4 LSTM layers, the first 2 with recurrent connections on the time axis and having 200 cells each while the last 2 with recurrent connections on the note axis and having 100 cells each. Adadelta (Zeiler, 2012) was used as the optimizer and the model was trained for 50,000 iterations with a batch size of 16 and cross-entropy used as a loss function as defined in the original paper. During training a new random batch of tracks is sampled every 10 iterations, and 1% of the training set is used for validation purposes during training. A dropout value of 0.5 is used in the LSTM cells to avoid over-fitting.

The AIRL discriminator consists of 2 layers of 64 feed-forward cells each. The generator is a Gaussian MLP Policy with 2 different neural networks, the first one is 2x64 layer MLP with a 44 cell output layer that outputs the mean for each possible note, while the second neural network is a 2x64 layer MLP with a 44 cell output layer that outputs the standard deviation. The batch size for training both the generator and the discriminator is 256, with Adam (Kingma & Ba, 2015) as the optimizer for both. The policy optimization is done using Trust Region Policy Optimization (TRPO) (Schulman et al., 2015), a policy gradient method effective for optimizing large nonlinear policies such as neural networks. The models were trained for 500 epochs doing 10 training iteration on both the generator and discriminator on each epoch.

Both the Q-network and the Target Q-Network have the same architecture as the Bi-axial LSTM and are initialized with the same weights. Training is done by using an epsilon-greedy exploration, with an initial exploration probability set to 1 that slowly decays to 0.1. The reward discount factor was set to 0.5. The Target Q-Network's weights are gradually updated to be the same as the Q-network with an update rate of 0.01. The model was trained for 120,000 iterations with sampling from the experience replay buffer happening every 12 iterations and the batch size set to 24, meaning 8 full trajectories with 120 time steps each are used to train every 60 iterations. The experience replay buffer has a maximum capacity of 200 full compositions of 120 time steps each, and after it is full the first composition is removed and the new composition is added, emulating the behavior of a queue. Adam is used as the optimizer just like in the RL Tuner Model. Three different models are trained with 3 different rewards, one with only the music theory reward, another with only the AIRL reward and another with the combination of both with the $c$ value of 0.5 giving equal weights to both rewards. We chose $c = 0.5$ because we wanted a reward that incorporated both types of rewards almost equally.

## 6 RESULTS

The following results show objective and subjective comparisons between 3 different models:

- Bi-axial Model: the base Bi-axial Model trained only with Deep Supervised Learning with no additional tuning. Examples of compositions: `https://drive.google.com/drive/folders/1ph3Nw5R5aIpSbeHm483zu2zRZgA-B_JC?usp=sharing`

- AIRL Model: pre-trained Bi-axial Model tuned using Deep Q-Learning with AIRL reward only. Examples of compositions: `https://drive.google.com/drive/folders/1AS729ejPoSFib2ipzsf6On0IAawcoKGp?usp=sharing`

- MT+AIRL Model: pre-trained Bi-axial Model tuned using Deep Q-Learning with a combination of both AIRL and music theory rewards. Examples of compositions: `https://drive.google.com/drive/folders/1MisDA3_F5EO_V5Mh9x1bU76cwLKcgidZ?usp=sharing`

A model trained with only a Music Theory Reward was considered for the comparison, however after training we realized the compositions created by this model tend to sound very similar. In other words, two different compositions generated by the model trained with only music theory reward will sound very similar to each other, showing a lack of creativity and diversity. Therefore, we decided not to include this model in the comparisons since it has no practical applications.

### 6.1 OBJECTIVE EVALUATION

We generate 2000 compositions with each of the 3 models, each composition starting with one of the different possible starting notes obtained from the training data. In order to objectively compare compositions generated by the different models we need to define some metrics that can describe the compositions. We use 6 metrics known to accurately describe the compositions pulled from previously published papers in the symbolic music generation field to compare the human composed tracks in the training data to a set of compositions generated by each of the models. The metrics used are ratio of polyphonicity (Dong & Yang, 2018), ratio of steps in scale (Guimaraes et al., 2017), ratio of unique chords (Akbari & Liang, 2018), ratio of chords repeated (Jaques et al., 2017), ratio of tonality (Guimaraes et al., 2017) and ratio of chords in repeated motif (Jaques et al., 2017). All of the metrics are ratios obtained by dividing the number of times an action satisfied a condition

|                               | Bi-axial     | AIRL         | MT+AIRL      |
|-------------------------------|--------------|--------------|--------------|
| Ratio polyphonicity           | (+)0.108861  | (-)0.040719  | (+)0.020159  |
| Ratio steps in scale          | (+)0.301088  | (+)0.092172  | (+)0.171911  |
| Ratio unique chords           | (+)0.540401  | (-)0.026132  | (+)0.078249  |
| Ratio chords repeated         | (-)0.157354  | (+)0.085684  | (-)0.031908  |
| Ratio tonality                | (+)0.022615  | (-)0.058328  | (+)0.006202  |
| Ratio chords in repeated motif| (-)0.275816  | (-)0.270690  | (-)0.264218  |

Table 1: Evaluation of Bi-axial, AIRL and MT+AIRL models against 6 objective metrics.

(for example in the case of ratio of polyphonicity, every time an action contains multiple notes in one timestep) over the total number of notes in a composition and have a value between 0 and 1. We calculate the values of these metrics for the human-composed songs in the training data in order to compare the models. We consider the closer the value of the metric gets to the training data the better, because that means the compositions are closer to human composed tracks.

Table 1 shows the difference between the metrics for each specific model and the values of the metrics for the human-composed tracks. The lower the value, the closer the metric is to the human composed track. We can observe in Table 1 that the AIRL Model has lower values in 5 out of the 6 metrics compared to the Bi-axial Model. The MT+AIRL Model shows an even greater improvement by having even lower values in all 6 metrics.

## 6.2 Subjective Evaluation

We conducted a User Study involving 26 participants via Google Forms in which the participants were asked to choose their preferred compositions between 2 options. Out of the 26 participants, 14 indicated having music theory knowledge or having experience playing an instrument. The participants choose between two compositions (each composition having a duration of 30 seconds) that have been generated by either a Human, Bi-axial Model, AIRL Model or MT+AIRL Model. In each survey every model is compared to every model, leading to a total of 6 comparisons between the 4 different models including human generated compositions. Each comparison is done with different tracks, so a total of 8 tracks per model are used in all the comparisons. A total of 156 comparisons were recorded; each model is involved in a total of 78 comparisons, 3 comparisons for each of the 26 participants. The objective is to force the users to select one composition over the other, to get a clear comparison of which model is preferred by the participants as a whole. We can observe the overall results of the user study and how many times each model was preferred (out of 78 comparisons) in Figure 3.

We can observe the human compositions are overall preferred over all other compositions as expected. However, there is no significant difference between the Human compositions and the MT+AIRL Model. Both models trained with RL are preferred over the base Bi-axial Model, with the MT+AIRL Model compositions being preferred by a larger margin. Tracks generated by the MT+AIRL Model were selected 46% of times when compared directly to human tracks, showing that a significant amount of people preferred tracks generated by the MT+AIRL Model over the human composed tracks. The AIRL Model shows an improvement in user preference when comparing it directly to the Bi-axial model. The Bi-axial model was only chosen on 38% of the responses when compared directly to the AIRL Model, while the AIRL Model compositions where chosen 62% of times. The MT+AIRL Model shows an even greater improvement over the Bi-axial Model, being selected an overwhelming 88% of times when directly compared against the Bi-axial Model.

## 7 Conclusions and Future Work

In this paper we propose an alternative approach to harmony generation, using three distinct training phases combining Deep Supervised Learning, Deep RL and IRL. A well designed reward function is required in order to use RL to learn a policy for music composition. In theory, AIRL is able to learn a reward function that allows RL to learn a policy to compose good music. However, due to the limited training data available, AIRL cannot easily learn a reward function that complies with

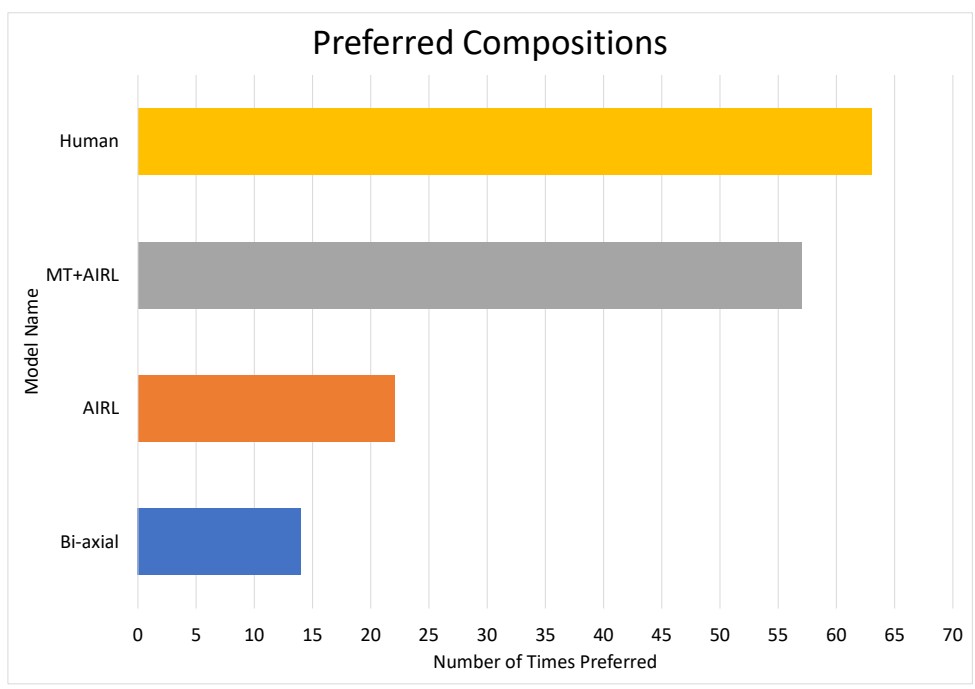

Figure 3: User study overall results

music theory all the time. On the other hand, a system based on music theory alone cannot generate a versatile and creative music piece. The proposed model, the model tuned with a combination of music theory and AIRL rewards (MT+AIRL Model), seems to be a compromise that can generate compositions that comply with music theory rules, keeping uniformity through the composition and not showing a fragmented style, while still showing some degree of creativity and variety. This model also has consistently the best results in both objective and subjective evaluation.

Models pre-trained with Deep Supervised Learning and later tuned using RL show better performance than a pre-trained model without RL tuning in multiple objective metrics that describe the characteristics of the compositions being generated. RL tuning with an AIRL reward function shows an improvement in 5 out of 6 objective metrics. Furthermore, when combining the AIRL reward with the supporting music theory reward, the model can be further improved, showing an even greater improvement in 6 out 6 metrics. As for subjective evaluation, the results of user studies show that participants prefer compositions generated by models tuned with RL over compositions generated by the model merely trained with Deep Supervised Learning. The tracks generated by the model trained with AIRL reward were chosen 62% of times over the tracks generated by the model only trained with Deep Supervised Learning. The tracks generated by the model tuned with a combination of AIRL and music theory rewards show an even greater difference in preference, as they were chosen over tracks generated by the model only trained with Deep Supervised Learning by 88% of participants. The tracks generated by the model tuned with a combination of AIRL and music theory rewards were preferred over human composed tracks by 46% of participants, showing the tracks generated by our model are very close to human composed tracks. We show that IRL is successful in capturing a reward function that can lead to better overall compositions that are preferred by humans and prove that RL is able to improve the performance of a base polyphonic music generation model.

Further investigation can focus on improving the IRL phase to include more information of the current compositions in the state representation. We used the previous chords in the composition to represent the state in the IRL algorithm, but it can be improved by including more information about the composition as a whole. We also believe there is room for improvement by experimenting with different models to use as pre-trained models.

REPRODUCIBILITY STATEMENT

A public link to the code used to train the models, inference the samples and calculate the objective metrics used in this paper will be included in the final version of the paper if accepted. To guarantee reproducibility the code contains the exact versions of external packages used and the Experiments section of the paper contains details of the hyperparameters used for training the models, as well as information on the datasets used and the preprocessing.

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
