# OpenReview forum: "Polyphonic Music Composition: An Adversarial Inverse Reinforcement Learning Approach"
_ICLR.cc/2022/Conference — ICLR 2022 Submitted_

### Official Review · Reviewer_n4MP · 2021-10-30

**Correctness:** 4
**Technical Novelty And Significance:** 3
**Empirical Novelty And Significance:** 3
**Recommendation:** 6
**Confidence:** 4

**Main Review:**

*Strengths*

The paper offers a nice combination of existing music-generation techniques, which, according to the metric used by the authors, outperforms any of these techniques alone

The proposed method generates polyphonic music, which is still rare

I like the overall idea of treating the problem of automated music generation as an RL task. Here the authors provide a proof of concept that such an approach is efficient for the task at hand.

Although this work combines the existing software with little-to-no alternations, the concepts offered here open up a lot of possibilities for finetuning the proposed model or using recent advances in relevant fields, such as offline RL.

The publications in automated music generation/processing are surprisingly rare and, I believe, every such attempt should be appreciated.

*Weaknesses*

The text could use more clarity when it comes to the methods. For example, to figure out the RL part of the model, I had to explore Figure 2 instead of reading the related portions of the text. Moving the first part of the Experiments section up in the text, renaming it to Methods, and appending it with details may help.

In the adversarial RL (AIRL) part of the algorithm, it is unclear whether the authors use LSTM in their network (from the writing it follows that they don’t). Conditioning the reward function on the latest chord only will prevent the model from encouraging long-term dependencies crucial for human-produced music. I do hope, however, that it’s a typo and the authors did include their LSTM in AIRL. If there is no LSTM in AIRL, adding it there will be an easy way to improve the model’s performance. Please clarify.

The authors present the samples of music generated by various algorithms, but not the baseline human-generated pieces which were the part of the comparison reported in the paper. While the music generated by the authors' pipeline sounds better than that produced by the baselines, it still sounds artificial and seems to lack temporal structure. In that light, it is unclear how it may have the same preference score as human-generated music. Perhaps, if the authors upload some samples of the music from their training set, the results of the subjective comparison will become more transparent.

*Specific comments*

Please clarify whether the “Q-network” and the “Target Q-network” have the same weights and are updated simultaneously (that is, their weights are always the same, but activations may differ). Please specify it in the text where the two networks are first mentioned.

Please clarify whether LSTMs are used in AIRL and, if there are no LSTMs in AIRL, please comment on the (in)consistency of the state definitions in AIRL and DQN and the resulting (un)transferability of the reward function.

Please provide a comprehensive description of AIRL in section 4.1., e.g. describe the exact relation between the discriminator network and the reward function. Please also provide the equations for that.

Please reformat the volunteers’ preference chart a 4x4 matrix/heatmap indicating, for every pair of the algorithms, which one was preferred and in what fraction of cases. Otherwise, in the current format, it's hard to interpret.

*Suggestions to the authors*

If there’s no LSTM in AIRL, please consider adding it for the following reasons:
1)	If the reward function, learned by the discriminator, is based on the last chord only, it does not reflect long-term dependencies, normally observed in music. Music can hardly be viewed as an MDP and not accounting for long-term dependencies would prevent the models from learning to generate human-like music.
2)	In the RL part of the model, the state (as in: the latest chord) is passed through an LSTM, so the *real* state, for which the Q-function is computed, accounts for long-range dependencies in time and is *not* the same with the state in AIRL, for which the reward function has been learned.

It is unclear whether there is any utility in pre-training the LSTM at all. Having an LSTM is critical for long-term dependencies in music, so it’s great that the authors have it in their model, but the objectives are different in pre-training (where the LSTM is trained to produce *actions*) and in finetuning (training to produce *Q-values*). I guess that the model would work just fine with no pre-training. If so, it will simplify the model. If not, the direct comparison of pre-trained vs. not pre-trained LSTMs would better substantiate the design choices.

Following up on the previous point, it seems to make more sense to apply the actor-critic framework here. Soft Actor-Critic (SAC), offering continuous control required by the authors, seems to be a good candidate algorithm here. It may solve a few problems at once:
1)	The objective for the LSTM part would be the same for pre-training and finetuning (as in: the probabilities of the *actions*); in the finetuning stage, the authors may simply add another head to the network computing the value functions for the states.
2)	It will solve an issue raised by the authors that all the Q-values are similar and the model tends to simply produce as many notes in a chord as possible – for which the authors have introduced a workaround. In SAC, the action probabilities are separate from the state values, and the aforementioned issue does not emerge.
With that said, I understand that the authors have limited time now, so this comment may be treated as a suggestion for future directions.


**Summary Of The Paper:**

This paper introduces a novel pipeline for generating polyphonic music. To generate music, the authors first pretrained a bi-axial LSTM (in time and pitch dimensions) on a corpus of piano rolls. To finetune the model’s performance, the authors then viewed music generation as an RL task and further trained their LSTM-based model to compute the values (Q-functions) of adding particular notes to the next chord. To specify a reward function for their RL approach, the authors used adversarial inverse RL (AIRL), where the reward function was approximated by the discriminator network. The blind evaluation of generated music by volunteers has indicated the volunteers’ preference towards human-generated and the proposed-model-generated music, but not to the music generated by its constituent baseline approaches.

**Summary Of The Review:**

The paper provides a novel approach to generating music and opens up numerous avenues for follow-up research. Impactful concepts and impressive results in this paper, alongside the general lack of publications on this topic, will make this paper a relevant contribution to the venue. The text, however, needs substantial edits to improve clarity and incorporate details of the proposed model – which, I hope, the authors will be able to do during the rebuttal period. I, therefore, tentatively recommend accepting the paper.

---

### Official Review · Reviewer_e8mJ · 2021-11-02

**Correctness:** 2
**Technical Novelty And Significance:** 1
**Empirical Novelty And Significance:** 2
**Recommendation:** 3
**Confidence:** 5

**Main Review:**

I find the framing of this paper troubling, starting with the first sentence:

"Automatic music composition usages can vary from continuous generation of copyright free music for use in media to inspiration tools for musicians."

It is not clear to me that music generated by an automated system is copyright free (if it was trained on copyrighted music). On the one hand, this point isn't central to the paper; on the other hand, this type of unsupported cavalier claim is representative of the writing in this paper. I note that there is not a single reference to the literature to support any of the claims made in the introduction to this paper.

Furthermore (and more importantly) I find the framing confusing. "[Music composition] can be viewed as generating a sequence of music chords, in which selecting each chord’s vector representation is equivalent to an action." From a precise musicological perspective, this is false: e.g. contrapuntal music is not constructed from chords. From a modeling perspective, it is common to model polyphonic music as a sequence of multi-hot vectors that can loosely be described as "chords." Perhaps this is what is meant but it is not clear, and the data representation and usage of the word "chord" remains confusing to me later in Section 4. The authors also regularly discuss "harmony" and "harmony composition" when I think they really mean full-blown polyphonic composition; this is confusing because harmonization (given a melody) is a different task.


Section 2 is far from a comprehensive discussion of music generation, and the particular choices of references are strange: for example, why cite WaveNet at all in a paper about symbolic music generation? If WaveNet is cited, why omit more recent examples of audio generation such as Jukebox?

The pattern of strong, unsupported claims is continued in Section 2: "when using GANs to generate music compositions directly, it tends to generate pieces of music too similar to the given illustrated examples that lack novelty and creativity." Furthermore, the focus on GAN's (which also appears in the paper's abstract) ignores that many music generation systems are not based on GAN's. I also find the claim of GAN popularity for "natural language Q/A fields" suspicious (and again there is no citation to support this claim).


The methodology in Section 4 is not clearly described. The authors describe the challenge of generating combinatorial combinations of notes (chords) in Section 4.1, but seem to claim that this is a continuous action space (it is not; it is a large, discrete action space). They also claim that "the state is represented by the previous chord," which seems to suggest that this is a Markov model with order 1; good music generation systems incorporate much more historical context. I think it is likely that continuous embeddings are being used to represent both actions and history, but this is an educated guess that isn't described well in the paper. What is the actual representation of the data that is being modeled? A sequence (time) x 44 piano-roll matrix, where time is discretized at some fixed resolution?


I find the empirical evaluation underwhelming. The baseline bi-axial results are not strong, so I don't find the improvements with AIRL or MT+AIRL very impressive. And the MT+AIRL results themselves do not seem particularly good to me compared to SOTA music generation systems. I would be very curious to see results using a stronger baseline model (e.g., Music Transformer). I wonder if the choice of dataset is also at issue here: my understanding is that is was constructed by ripping piano tracks out of Lakh Midi compositions. I would be interested to hear ground truth samples from this dataset (maybe my expectations are too high for the generations, if the ground truth isn't very impressive). A solo piano dataset such as MAESTRO might be a better choice. Finally, the fact that MT+AIRL performs well quantitatively seems unsurprising, given that the MT rewards (Section 4.2) appear to explicitly promote several of these metrics (Goodhart's law).

**Summary Of The Paper:**

This work proposes an approach to polyphonic music composition based on reinforcement learning. The approach constructs a reward function based on inverse reinforcement learning (IRL) using human demonstrations, in conjunction with a hand-crafted, music-theoretic reward. Empirical evaluation compares the approach based in IRL to a baseline LSTM density estimator.

**Summary Of The Review:**

The main contribution of this paper is an adaptation of inverse reinforcement learning techniques to the music generation domain. But the discussion of the music domain is confusing (from both a musicological and technical modeling perspective) and so much of the potential value of this adaptation is lost. The empirical results are also underwhelming.

---

### Official Review · Reviewer_N31m · 2021-11-03

**Correctness:** 3
**Technical Novelty And Significance:** 3
**Empirical Novelty And Significance:** 4
**Recommendation:** 5
**Confidence:** 4

**Main Review:**

The application of AIRL to learn a reward function for music generation is the main contribution of this paper. This reward function leads to a huge improvement in generation quality of the base RNN model as is evident from the provided music samples. Combined with a music theoretic reward, the model generates okay sounding piano music (lack of structure, repetitions, cadence), which is not an easy task.

That being said, there are a few issues in this paper.
- First, the writing can be organized a bit better. I started off with the idea that this paper is only focusing on chord generation since that seems to be the focus in the first couple of pages. Then I doubt myself, and finally in section 5 it is clear that the piano-rolls are not only for chord sequences, but full piano performances/tracks with harmony and melody. The strong emphasis on harmony also led to this confusion.
- Please consider changing the structure to mention the data representation before section 4.1. This also helps with lack of context when referring to MIDI or piano rolls or why you only have 44 notes in the piano roll, which are all talked about in section 4.1 without any context.
- In section 3.3 and 4.1 you mention this special structure in the AIRL model which enables the reward function to be extracted from the discriminator. Perhaps at one of these places you can elaborate on the special structure.
- The evaluation section only consists of ablation studies, i.e. comparison between the base LSTM, the AIRL LSTM, and AIRL+MT LSTM. There needs to be some comparison with other popular polyphonic music generation model such as MuseGAN/Music Transformer to put the quality of music generated into context. Also, it would be nice to include samples of the human generated music that were used in the subjective evaluation as well.
- The related work section does not mention some of the more recent work in music generation. The latest paper mentioned is the music transformer paper from 2018. There have been a lot more recent works focusing on polyphonic and piano music generation. Please consider discussing those works as well.
- You mention that the model tuned with only the music theory reward generates samples that sound very similar to each other. That sounds like possibly an implementation or training problem since this setup is exactly how RL-tuner is setup, unless I am mistaken (except the polyphony). Samples for this would also be appreciated.

Typos, suggestions, etc:
- unpleasing -> unpleasant
- Usages -> applications
- Only manually and subjectively assigning: I am not sure if this is ever actually done in music generation models. This sounds like a human-in-the-loop learning framework. Your sentence makes sense even without this phrase.
- IRl -> IRL
- Lahk -> Lakh


**Summary Of The Paper:**

This paper presents a method for polyphonic piano-based symbolic music generation based on previous work on RL-tuned recurrent networks. This paper adds an adversarial inverse reinforcement learning (AIRL) step to estimate a reward function which is used in tandem to music theoretic rewards during the Q-network tuning. The proposed model performs better than the untuned LSTM-based model on both subjective and objective metrics.

**Summary Of The Review:**

The paper proposes a novel approach to tuning an LSTM-based music generation model using a learned reward function in addition to music theory-based rewards. Adversarial inverse RL is used to that end. This addition leads to an improvement in the generated music. However, the quality of the generated music does not hold up to other recent work for polyphonic music generation. There are a few other minor issues in the paper listed above. Thus I rate this paper marginally below acceptance threshold.

---

### Decision · Program_Chairs · 2022-01-20

**Decision:**

Reject

**Comment:**

This work proposes a system for generating piano music (in the symbolic domain) using a learned reward function. Reviewers raised concerns about the organisation of the paper, clarity of writing, a lack of experimental comparison with previously published approaches (and the quality of the baseline), several unsubstantiated claims, and some missing related work. Unfortunately no attempt was made to address these issues.